# Current Evidence, Limitations and Future Challenges of Survival Prediction for Glioblastoma Based on Advanced Noninvasive Methods: A Narrative Review

**DOI:** 10.3390/medicina58121746

**Published:** 2022-11-29

**Authors:** Sergio García-García, Manuel García-Galindo, Ignacio Arrese, Rosario Sarabia, Santiago Cepeda

**Affiliations:** 1Department of Neurosurgery, University Hospital Río Hortega, Dulzaina 2, 47012 Valladolid, Spain; 2Faculty of Medicine, University of Valladolid, Avenida Ramón y Cajal 7, 47003 Valladolid, Spain

**Keywords:** glioblastoma, glioma, high-grade glioma, radiomics, artificial intelligence, deep learning, machine learning, survival

## Abstract

*Background and Objectives*: Survival estimation for patients diagnosed with Glioblastoma (GBM) is an important information to consider in patient management and communication. Despite some known risk factors, survival estimation remains a major challenge. Novel non-invasive technologies such as radiomics and artificial intelligence (AI) have been implemented to increase the accuracy of these predictions. In this article, we reviewed and discussed the most significant available research on survival estimation for GBM through advanced non-invasive methods. *Materials and Methods*: PubMed database was queried for articles reporting on survival prognosis for GBM through advanced image and data management methods. Articles including in their title or abstract the following terms were initially screened: ((glioma) AND (survival)) AND ((artificial intelligence) OR (radiomics)). Exclusively English full-text articles, reporting on humans, published as of 1 September 2022 were considered. Articles not reporting on overall survival, evaluating the effects of new therapies or including other tumors were excluded. Research with a radiomics-based methodology were evaluated using the radiomics quality score (RQS). *Results*: 382 articles were identified. After applying the inclusion criteria, 46 articles remained for further analysis. These articles were thoroughly assessed, summarized and discussed. The results of the RQS revealed some of the limitations of current radiomics investigation on this field. Limitations of analyzed studies included data availability, patient selection and heterogeneity of methodologies. Future challenges on this field are increasing data availability, improving the general understanding of how AI handles data and establishing solid correlations between image features and tumor’s biology. *Conclusions*: Radiomics and AI methods of data processing offer a new paradigm of possibilities to tackle the question of survival prognosis in GBM.

## 1. Introduction

Glioblastoma (GBM), with an incidence of 3.5/100.000 population, is the most common malignant primary neoplasm of the brain in adults [1]. Its fatal prognosis has scarcely improved over the last decades despite intensive research in the field [1]. Some clinical, surgical and radiological features are known independent predictors of survival [2]. However, accurate survival prediction is a key challenge for patients, relatives and physicians in their search for precision medicine strategies to tackle the burden of this devastating tumor [3].

A radiology-based approach to prognosis prediction has gained momentum in recent years fostered by the development of advanced tools to manage an immense quantity of data and thanks to its noninvasiveness, radiomics, a science based on quantitative data mining from medical images, has been extensively used in oncology [4]. Texture analysis, a branch of radiomics, exploits the information concealed in voxels and pixels and provides a quantitative assessment of images that might serve as a virtual biopsy [5]. Thus, different imaging modalities, segmentation algorithms and texture features have successfully contributed to supporting tumor diagnosis, molecular profile estimation, treatment response evaluation and overall survival (OS) prediction in neurooncology [6,7,8].

Handling such an enormous quantity of complex data as derived from radiomic analysis requires specific methods to obtain useful results and interpretable information. Thus, artificial intelligence (AI) methods have been developed and applied to this novel prognostic approach [9]. Machine learning (ML), a discipline within AI, through training datasets on ground truth labels has allowed us to obtain algorithms that can execute complex tasks such as tumor segmentation, tumor grading, molecular classification and survival prognosis [10]. ML may follow a supervised (e.g., *logistic regression*, *support vector machine (SVM)*, *random forest (RF)*, *naïve Bayesian networks*, *decision trees (DT)*) or unsupervised (e.g., *K-means cluster*) workflow. In a different way, deep learning (DL), a subclass of ML, does not require human intervention or ground truth labels to learn. Instead, DL, mainly present in different forms of neural networks (NN), has been successfully applied to tasks such as survival estimation, tumor segmentation, and estimation of glioma molecular subtypes [6,11,12,13,14,15]. NN learns on its own from previous fails and successful associations and can make more complex correlations. However, DL needs higher volumes of data and requires extensive computational time for training. In all, different AI strategies might be used to perform similar functions, which in the case of survival prognosis often consists of correctly classifying patients into long and short survivors, whether by means of SVM, RF, DT or naïve Bayesian classifiers, for instance, in the case of ML; or NN in the case of DL.

In this review, we present an update of the current evidence on advanced statistics, radiomics and data processing methods for the accurate survival prognosis of patients suffering from high-grade gliomas. The initial attempt to conduct a systematic review or meta-analysis was soon quit given the enormous heterogeneity of methods, the frequent involvement of the same set of patients from public datasets and the lack of consistency in results reporting. Instead, we sought to provide a comprehensive and thorough review of advanced noninvasive methods for survival estimation in GBM, while we discuss present challenges and future perspectives in this field.

## 2. Materials and Methods

A systematic search of PubMed was performed for articles reporting on the survival prognosis for GBM, involving in their methodology a noninvasive advanced radiological approach including radiomics and/or AI.

The query was built for the following terms to be present on the Title or abstract: ((glioma) AND (survival)) AND ((artificial intelligence) OR (radiomics)). Only English full-text articles, reporting on human subjects, published (even *ahead of print*) as of 1 September 2022 were considered. Each manuscript was independently reviewed by two authors (S. G-G and S.C.). Articles not reporting on overall survival, including other tumors than GBM, or redundant articles were excluded. Articles assessing the effect of novel therapies on survival were also excluded. In the case of an investigation conducted by the same scientific team with similar methods applied to an updated, previously published dataset, the last manuscript was considered. In case of disagreement on the need to include a given study, a consensus was reached after substantiated discussion held by the signing authors.

Studies involving a methodology based on radiomics were assessed according to the radiomics quality score (RQS) [16]. The RQS is a sixteen-item scale that evaluates the methodology and reporting of an investigation to improve the consistency of radiomics studies, increase its reproducibility and enhance scientific soundness in this field [16].

## 3. Results

The search resulted in 382 articles with the terms glioma, survival, AI and radiomics, whether in their title or abstract. After applying the inclusion criteria, 311 articles were excluded. Seventy-one full-text articles were thoroughly reviewed, of which 46 articles were ultimately included (Figure 1). The main results of these articles are summarized in Table 1. Similarly, the results of the RQS of articles, whose methodology was based on radiomics, are presented in Table 2. The average and median scores of the RQS for the analysed studies were 10.5 (29%) and 11 (31%), respectively.

## 4. Discussion

Predicting life expectancy for patients diagnosed with GBM is a major challenge. Many variables might influence the prognosis, and despite numerous efforts and approaches, uncertainty often erodes patients’ and physicians’ will to face and overcome the fateful outlook that this disease entails. New advanced techniques to process images and data have proven their ability to noninvasively mine and display the information shielded in the infinite number of pixels and voxels that compose multimodal neuroimages. Initially, we conducted a meta-analysis or systematic review of the main evidence on survival prognosis through these advanced methods, such as radiomics and AI. However, we soon realized that the tremendous variability of methods, the lack of consistency in reporting the results and the recurrent use of public datasets of the same patients hampered successful fulfillment of this task and rendered it unmanageable. Instead, we present a thorough review of the key available evidence regarding this critical issue while we discuss the main existing limitations, the actual room for improvement and the new horizons and challenges that future research should address.

Conventional morphological MRI features contain valuable prognostic information. As proven by Molina et al., a simple model based on age and morphological features, without texture data, could outperform more complex models in GBM [32] prognosis prediction. Nonetheless, morphologic information might be a loose term in which many features fit. In an effort to unify the common characteristics gliomas display on MRI studies and to harmonize the vocabulary in which we refer to them, Visually Accessible Rembrandt Images (VASARI) were developed [57]. This list of 30 features has been demonstrated to be useful for predicting OS in GBM [58]. In addition, VASARI, which does not consider texture information, has been successfully integrated into ML models based on texture data [56]. However, in a study by Ruan et al., only *cortical involvement* was strongly associated with poor outcomes. Indeed, VASARI features did not increase the predictive accuracy of the RF model based on radiomic features when combined with them.

As shown in Table 1, there are several examples of predictive algorithms based on texture features extracted from conventional MRI sequences. Studies vary widely in terms of image preprocessing, segmentation methods, number of features and ML classifiers (Figure 2). Different strategies and approaches have been performed with a few outstanding examples, such as those of Ruan et al.; Cepeda et al.; Lu et al.; and Sanghani et al., in whose studies based on conventional MRI sequences and ML algorithms, accurate prognostic classification exceeded 90% [29,52,55,56]. Moreover, advanced MRI sequences can also be employed for prognostic purposes. Thus, some authors have implemented functional resting-state MRI and DTI (Diffusion Tensor Imaging) studies to build structural and connectivity networks, extract features and process them with ML and DL algorithms [19,50,59].

An interesting conceptual approach was implemented by Lee et al., who used relative texture features from perfusion maps to predict the survival status at 12 months [22]. Features extracted from enhancing and nonenhancing regions of the tumor were computed to provide these relative features. In addition, kinetic features calculated from the gadolinium concentration time-series of perfusion data in both regions were also extracted [22]. Nonetheless, relative texture features displayed a higher impact on prognosis than kinetic features [22].

Delta-radiomics, which consists of features extracted at different time points, provides information about how radiomic features change over time. In a small cohort, Chang et al. demonstrated the strong potential of delta-radiomics combined with a DL algorithm [39]. It is still unknown which time lapses might be more informative depending on the studied outcome variable. Regardless, delta features seem to provide higher predictive information than one-time features, which makes it even more logical for survival prognosis.

ML methods may also generate different radiomic profiles of gliomas that could potentially translate their underlying biological features. Itakura et al. and Rathore et al. classified gliomas into three clusters with different survival prognoses using different methodologies [60,61]. Remarkably, both teams reported three different radiomic profiles that actually shared some characteristics [9]. Specifically, rim-enhancing subtypes granted better prognosis in both reports. Indeed, tumor subgroups were associated with molecular subtypes, location and genetic mutations [60,61]. These findings, which require further validation, support the idea of radiomics as a virtual biopsy, turning the corner to more personalized and precise communication with patients and relatives.

Deep learning, as a branch of AI and considered a fully machine learning method able to train on its own without human intervention, has consistently defeated rivals in image recognition competitions such as ImageNet. The increasing amount, complexity, types and availability of data partially explains the swift direction towards DL for medical applications, as DL does not strictly require a human-driven refining of data.

Moradman et al. demonstrated the superior ability of DL to establish complex correlations between multiple clinical, biological and therapeutic variables, and survival in patients diagnosed with glioblastoma [11]. A feed-forward NN offered higher accuracy on survival prognosis than the random survival forest and Cox proportional hazard regression models [11].

Obviating the tasks that texture analysis involves, Ben Ahmed et al. built a convolutional NN based on MRI snapshots that outperformed the accuracy of the best-known predictor by 6% [14]. Using nonlabelled, nonsegmented snapshots offers a fast and low-cost way to feed a DL algorithm [14]. Nonetheless, DL might also be used as a method to mine images. With a hybrid approach, Nie et al. applied a DL method to automatically extract features that would otherwise be difficult to design [34]. These DL features together with key demographic and tumor-related features allowed us to stratify patients into long and short survival groups through an SVM classifier with 90.5% accuracy [34]. DL also supports the inclusion of advanced MRI sequences, as demonstrated by Yan et al., who compared a clinical nomogram with a DL signature based on DTI data [62]. Yan et al. obtained better results with the DL signature, achieving a C-index of 0.9 on an external validation dataset [62]. In addition, the authors suggested the existence of an association of DL features with biological pathways involved in glioma development (synaptic transmission, activation of AMPA receptors, axon guidance, calcium transport, etc.) [62].

### 4.1. Future Challenges

#### 4.1.1. Data Availability

AI training requires a large, high-quality dataset to build robust algorithms. Creating such datasets is costly and time consuming and demands that professionals shift from care provision to data production. This burden is especially problematic in rare diseases such as GBM. Therefore, a culture of data sharing is needed. Cooperative efforts, such as The Cancer Imaging Archive or the Ivy Glioblastoma Atlas project, have contributed to increasing data availability. Additionally, data could be improved by the harmonization of image acquisition protocols across institutions. Automatic data acquisition, often used by AI in other fields, clashes with the need to preserve the confidentiality of medical data [63].

#### 4.1.2. Opening the Black Box

AI algorithms are built from associations that are not fully disclosed by the algorithm itself. Therefore, drawing conclusions between radiomic features and glioma characteristics might be misleading since their relationships are unknown, and predictive models might be based on variables derived from similar features that might be overrepresented [64]. Indeed, understanding the underlying mechanisms by which biology translates into radiomic features is a classic concern and matter of current investigations. Nonetheless, recent advances, such as principal component analysis and saliency maps, have relieved these concerns by unveiling part of the structure of AI algorithms [65]. Radiomic features might be the fine manifestation of molecular phenotypes in grayscale images [20].

#### 4.1.3. Humanizing AI

When AI is implemented to fulfil a given task, human vs. machine approaches are often used to elucidate who can better perform it. Modern AI applications to the medical field have suggested the benefits of *human-in-the-loop* strategies to overcome the unique challenges medicine poses to AI. In expert augmented machine learning, researchers combined the knowledge of experts to solve specific problems where AI algorithms might fail the most [66]. Thus, the quality of training data might be notably improved by integrating the information that specialists base their decisions on. Similarly, in *active learning*, key data are obtained from the expert by the algorithm itself to increase the quality of the training dataset or enhance the ability of the algorithm to extract useful information [67,68].

#### 4.1.4. Integrating AI into Clinical Practice

The ultimate goal of research in medicine is transferring the lessons learned in the laboratory to clinical practice. The topic covered in this review is not an exception. The reports presented herein are commendable efforts to find key features and methods to improve patients’ prognostic estimation. However, the methodology that most of these investigations involve is extremely time-consuming and makes it inefficient for daily implementation in a clinical context. The next paramount advancement in this field, beyond increasing accuracy or simplifying the workflow, will be the production of an open source, easily integrable and precise AI algorithm that requires simple or null intervention of the physician for prognostic estimation from multimodal MRI studies.

### 4.2. Limitations

The major limitations of previous publications can be summarized in the following sections:Patient selection: In most published articles, patients were included without considering the extent of resection, which is one of the main factors associated with overall survival. Therefore, if the intention is to use the imaging characteristics independently to predict the outcome, it is necessary either to include only patients with gross total resection or perhaps to introduce in the model a variable through which the degree of resectability of the tumor can be quantified [55].Image preprocessing and data extraction: There is significant variability in the methods employed to preprocess MRI images and in the parameters used to extract radiomic features. This pitfall explains the differences in the results obtained on the same patient dataset (such as the TCIA patient cohort) [47,49]. Therefore, the lack of details about the preprocessing pipeline used by the different authors limits the reproducibility of their results [11,35,44,54].Classification task vs. survival regression: There are discrepancies in how different authors approach the challenge of predicting survival in GBM. On the one hand, some studies attempt to carry out a survival analysis, in which the relationship between the radiomic variables and survival in days is expressed by the Harrell index or the hazard ratio [6,18,35,37]. On the other hand, there are works in which a classification task has been carried out to create survival groups. The latter methodology is much easier to interpret and has a clinical orientation [6,19,23]. However, the cut-off point for establishing survival groups is entirely arbitrary in various publications [19]. For example, it does not seem helpful to define a short-term survivor as one who does not exceed ten months of life when the overall median survival is 15 months. Therefore, unifying the criteria for short- and long-term survival definitions in this neoplasm is essential.Lack of validation in multi-institutional data: Although there are studies with promising results, the lack of validation in a multicenter cohort seriously limits the application of predictions in a clinical setting [55]. One of the challenges of models based on radiomic features is to find a set of stable and reproducible features so that they can be used regardless of artifacts produced during image acquisition, MRI acquisition protocols, and scanner manufacturers.

## 5. Conclusions

Advanced image analysis and data processing methods have gained momentum over the last decade. Methods such as radiomics, texture analysis, ML and DL have been successfully implemented to provide an accurate survival estimation and risk factor identification for patients diagnosed with GBM. The wide variety of available approaches prevents unifying methods and drawing consistent conclusions from reported results. However, despite its limitations, the existing symbiosis between radiomics and AI represents a robust approach to build evidence and address unanswered questions in neuro-oncology. In fact, AI is no longer a matter of future but a living, vibrant and powerful reality.

## Figures and Tables

**Figure 1 medicina-58-01746-f001:**
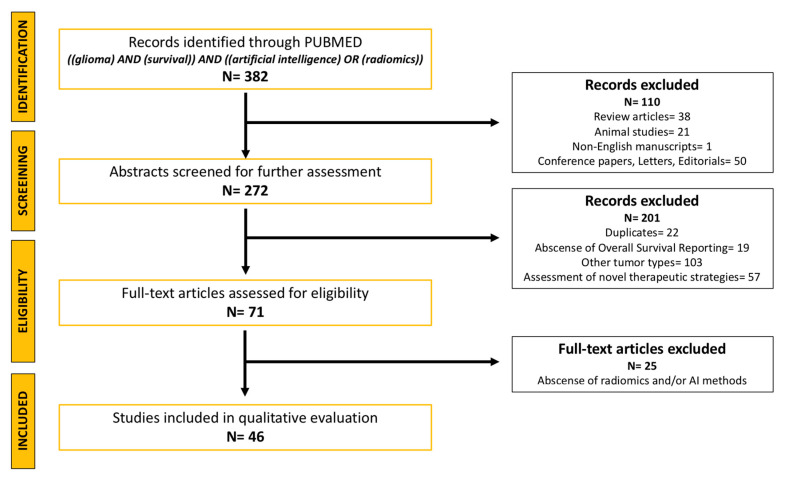
Flowchart depicting the identification, screening and inclusion of studies in the present review.

**Figure 2 medicina-58-01746-f002:**
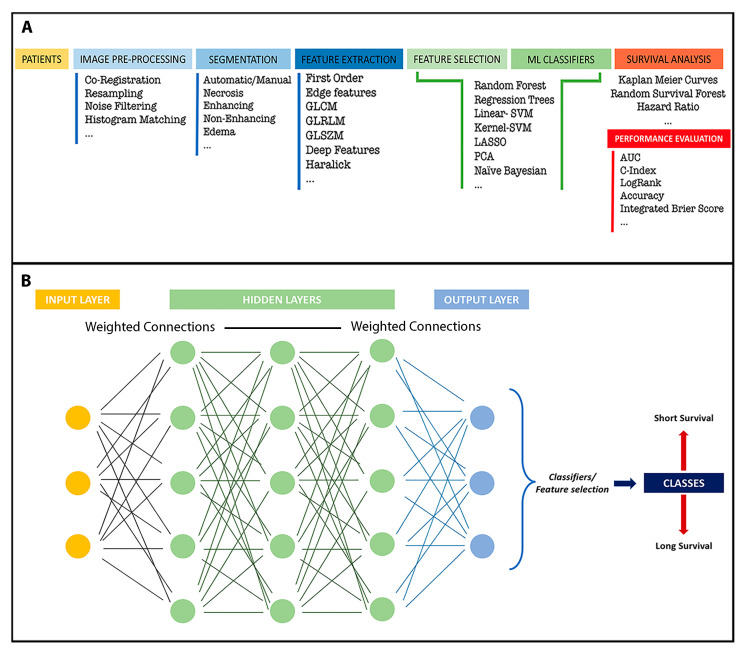
(**A**) Diagram depicting an example of the conventional process for studies implementing radiomics and machine learning algorithms. (**B**) Diagram of an example of a survival prediction investigation based on deep learning.

**Table 1 medicina-58-01746-t001:** Summary of the studies analyzed in this review.

	Author Year	N	Cases from Public Database *	MRI Sequence Radiomics Analysis	Segmentation Method (Labels)	Image Preprocessing	F. Extraction Software	N of F.	Feature Type	Feature Selection/ ML Classifier	Validation Method	Model Performance
1	Yang [17] 2015	82	Yes TCIA	T1C FLAIR	Manual Enhancing tumor Whole tumor	Intensity Normalization Re-Slicing	MATLAB	976	SFTA, GLRLM, Local Binary Patterns, Histogram of oriented gradients, Haralick	RF	Out-Bag Validation	SFTA T1C AUC = 0.69
2	Chaddad [18] 2016	40	Yes TCIA	T1 FLAIR	Manual Enhancing Tumor Necrosis Edema	Co-Registration Intensity Normalization	MATLAB	22	GLCM	DA, NB, DT, SVM	LOOCV	AUC = 0.793 Phenotypes with KM significantly different
3	Liu [19] 2016	68	No	RS-F- MRI DTI	Automatic Anatomical Labelling	RS-F-MRI: SPM8 and DPARSF DTI: FSL and PANDA	GRETNA Toolbox for Connectomics	2797	Functional and Structural Networks, Clinical	SVM	No	Accuracy = 75%
4	Macyszyn [20] 2016	134	No	T1C T1 T2 T2FLAIR DTI DS	Automatic Enhancing Tumor Non/Enhancing Tumor Edema Ventricles	Co-Registration	N/A	216	First Order, Tumor Location, GLISTR Outputs, Intensities	SVM	10-FoldCV VD = 29	Retrospective Accuracy = 77.14% Prospective Accuracy = 79.17%
5	Kickingereder [21] 2016	119	No	T1 T1C FLAIR DWI DS-C	Semiautomatic, Enhancing Tumor Non/Enhancing Tumor	Co-Registration N4 bias Correction Intensity Normalization	MITK	12,190	First order, volumetric, Wavelet, Haralick, GLCM, GLRLM,	SPCA	VD = 40	C-index = 0.61 HR = 3.45 KM
6	Lee [22] 2016	24	Yes TCIA	DS-C	Manual Enhancing tumor Nonenhancing tumor Normal WM	Co-Registration Normalization	MATLAB	18	First order, GLCM, Haralick	Univariate Analyisis	No	AUC = 0.83 HR = 0.019 KM
7	Ingrisch [23] 2017	66	No	T1C	Semiautomatic Whole Tumor	Resampling Normalization	Python	208	First order, Haralick, Parameter-free Threshold Adjacency Statistics	Minimal Depth, RF	10-FoldCV	C-index = 0.67 HR = 1.04 KM
8	Liu [24] 2017	133	Yes TCIA	T1C	Manual Enhancing Tumor	Resampling	MATLAB	56	First order, GLCM, GLRLM	RFE-SVM	10-FoldCV	AUC = 0.81 Accuracy = 78% KM
9	Li [25] 2017	92	Yes TCIA = 60	T1 T1C FLAIR T2	Automatic Enhancing Tumor Non/Enhancing Tumor Necrosis Edema	N4 bias correction, skull stripping, resampling, co-registration, histogram matching	MATLAB	45,792	First order, GLCM, GLRLM, GLSZM, NGTDM	LASSO	VD = 32	C-index = 0.71 HR = 3.29 KM
10	Liu [24] 2017	133	Yes TCGA	T1C	Manual Whole Tumor	Resampling	MATLAB	56	GLCM, GLRLM, Histogram	SVM	No	Accuracy = 78.2% AUC = 0.8104
11	Lao [15] 2017	112	Yes TCIA = 75	T1 T1C FLAIR T2	Manual Necrosis Enhancing tumor Edema	N4 Bias correction Resampling Co-Registration Histogram matching	MATLAB	99,707	First order, GLSM, GLRLM, GLSZM, NGTDM, Deep features	LASSO	VD = 37	C-index = 0.71 HR = 5.13 KM Nomogram
12	Prasanna [26] 2017	65	Yes TCIA	T1C FLAIR T2	Manual Enhancing Tumor Necrosis Edema	Co-Registration Insensity Normalization Bias Field Correction	MATLAB	402	Haralick, Laws features, Histogram of oriented gradients, Laplacian pyramids	mRMR, RF	3-FoldCV	KM C-index = 0.70
13	Kickingereder [27] 2018	181	No	T1 T1C FLAIR T2	Semiautomatic, Enhancing Tumor Nonenhancing tumor Necrosis	Intensity Normalization Coregistration	MITK	1043	First order, shape, GLCM, GLRLM, GLSZM	LASSO	VD = 61	HR = 2.72
14	Bae [28] 2018	217	No	T1C FLAIR T2 DTI	Manual. Necrosis Enhancing Tumor Non/Enhancing Tumor	Co-Registration N4-Bias Correction Normalization	Python	796	GLCM, GLRLM, GLSZM	VHA, RSF	VD = 54	AUC = 0.652 KM
15	Sanghani [29] 2018	163	Yes BRATS	T1 T1C FLAIR T2	Manual Enhancing Tumor Non/Enhancing Tumor Edema	Co-Registration Resampling	Python	2200	Volumetric, Shape, First order, GLCM, Gabor texture	RFE-SVM	5-FoldCV	Accuracy = 98.7%
16	Chaddad [30] 2018	40	Yes TCIA	T1 FLAIR	Manual Enhancing Tumor Non/Enhancing Tumor Necrosis Edema	Co-Registration Resampling Intensity Normalization	MATLAB	9	Texture features based on LOG filter	RF	5-FoldCV	AUC = 0.85
17	Liu [31] 2018	119	Yes	T1 T1C FLAIR T2	Manual Enhancing tumor	Co-Registration Resampling	MATLAB	54	First order, GLCM, GLRLM	SVM-RFE	No	T1C AUC = 0.79, Accuracy = 80.67% KM
18	Molina-Garcia [32] 2019	404	Yes TCIA	T1C	Manual Enhancing Tumor Necrotic Core	No	MATLAB	44	First Order, GLRLM, GLCM	NN SVM RT	VD = 93	C-Index = 0.817 (Optimal Linear Prognosis Model)
19	Tan [33] 2019	147	Yes TCIA = 112	T1C FLAIR	Manual Whole tumor Edema Contralateral WM	Co-Registration N4 Bias Correction Resampling Intensity Normalization	MATLAB	1456		LASSO	VD = 35	Radiomics C-index = 0.71 HR = 2.18 Nomogram C-Index = 0.76
20	Nie [34] 2019	93	No	T1C DTI RS-F-MRI	Manual Whole Tumor	Co-Registration	N/A	2048	CNN supervised	CNN SVM	10-FoldCV VD =25	Accuracy = 90.46% (VD = 88%)
21	Choi [35] 2019	114	Yes TCIA = 53	T2	Manual Peritumoral	N/A	Python	106	First Order, GLCM, GLRLM, GLSZM	No	VD = 34	C-index 0.659 KM
22	Chen [36] 2019	127	Yes TCIA	T1C	Manual Enhancing tumor	Insensity Normalization	MATLAB	3824	First order, Shape, GLCM, GLRLM	mRMR	N/A	HR = 3.65 AUC = 0.82 KM
23	Sasaki [37] 2019	182	No	T1 T1C T2	Manual Enhancing tumor Whole tumor	Co-Registration Intensity Normalization	MATLAB	489	First order, GLCM, GLRLM, shape	SPCA, LASSO	10-FoldCV	HR = 1.62 High and Low risk Log Rank Test *p* = 0.004
24	Um [38] 2019	161	Yes TCIA	T1 T1C FLAIR	Semiautomatic Whole tumor	Co-Registration Rescaling Bias field Correction Histogram Matching Resampling	CERR	420	First order, Edge features (LoG, Sober, Gabor, Wavelet), GLCM, GLSZM, Haralick	LASSO	VD = 47	HR = 3.61 KM
25	Chang [39] 2019	12	No	T1 T2FLAIR Pretreatment Posttreatment1 Posttreatment2	Manual Whole Tumor	Co-Registration	MATLAB	61	GLCM, GLDM, GLRLM, GLSZM, Delta Radiomics	RF, Linear- SVM, Kernel-SVM, NN, NB, LR	No	AUC = 0.889 Best Result: RF with SVM and NN with Delta Radiomics
26	Tixier [40] 2019	159	Yes TCIA = 47	T1 T1C FLAIR	Semiautomatic Whole tumor	Co-Registration Gabor Filtering Binning	CERR	286	First order, GLCM, GLSZM, Gabor	LASSO	VD = 61	KM
27	Shboul [41] 2019	224	Yes BRATS	T1 T1C FLAIR T2	Automatic Whole tumor Edema Necrosis Enhancing Tumor	Co-Registration Bias Correction Normalization	N/A	31,000	Texture, Euler, Histogram	Univariate, RFS, RF, XGBoost	VD = 61 LOOCV	Accuracy = 73% VD-Accuracy = 68%
28	Chaddad [42] 2019	200	Yes TCIA = 71	T1C, FLAIR	Manual Whole tumor	Resampling	MATLAB	45	First order, GLCM, NGTDM, GLSZM	No	VD = 100	AUC = 0.752 KM
29	Kim [43] 2019	83	No	T1 T1C FLAIR T2 DTI DS-C	Semiautomatic Enhancing Tumor Non/Enhancing Tumor	Co-Registration Intensity Normalization Resampling	MATLAB	6472	First order, Wavelet, GLCM, GLRLM	LASSO	10-FoldCV	DTI Radiomics AUC = 0.70 C-index 0.63 DS-C AUC = 0.76 C-index = 0.55
30	Liao [44] 2019	137	Yes TCIA	FLAIR	Manual	N/A	Python	72	First order, GLCM, GLSZM, GLRLM, NGTDM, GLDM	GBDT, SVM, kNN	VD = 41	Accuracy = 81% Short survival AUC = 0.79 Long survival AUC = 0.81
31	Osman [45] 2019	163	Yes, BRATS	T1 T1C FLAIR T2	Manual Enhancing Tumor Non/Enhancing Tumor Edema	Co-Registration Smoothing Interpolation Intensity Normalization Intensisty Rescaling	MATLAB	147	First order, GLCM, Histogram of oriented gradients, Local Binary Pattern.	LASSO, SVM, kNN, DA	VD = 54	Accuracy = 57.8% Short survival AUC = 0.81 Median survival AUC = 0.47 Long survival AUC = 0.72
32	Chaddad [46] 2019	73	Yes TCIA	T1C FLAIR	Manual Enhancing Tumor Necrosis Edema	Co-Registration Resampling Intensity Normalization	MATLAB	11	JIM, GLCM	SpCoR RF	LOOCV	JIM features: HR = 1.88 AUC = 0.776
33	Zhang 2019 [47]	105	Yes TCIA	T1 T1C FLAIR T2	Manual FLAIR Signal Enhancing Tumor Necrosis Edema	Co-Registration Resampling Collewet Normalization	MATLAB	4000	First Order, GLCM, GLRLM, GLSZM, Wavelet	LASSO LR	VD = 35	C-Index = 0.94 Nomogram
34	Han [6] 2020	178	Yes TCIA = 128	T1C	Manual Whole Tumor	Normalization Gray-Level Quantization Resampling	MATLAB (radiomics) CNN (Keras-TensorFlow) Elastic Net/Cox (R)	8540	First order, Nontexture, Histogram, GLCM, GLRLM, GLSZM, NGTDM Deep features(CNN)	MAD, C-Index, PearsonC	No	Long Rank Test Long/Short Survival *p* < 0.001 (HR = 3.26)
35	Zhang [48] 2020	104	Yes TCIA	T1 T1C FLAIR T2	Manual Whole Tumor Tumor subregions	Co-Registration Resampling Normalization	MATLAB	180	First Order, GLCM, GLRLM, GLSZM	Multiple Instance Learning, SVM	VD = 33	Accuracy = 87.9% Sensitivity = 85.7% Specificity = 89.4%
36	Suter [49] 2020	109	Yes TCIA = 76	T1 T1C FLAIR T2	Automatic Enhancing Tumor Non/Enhancing Tumor Necrosis Edema	Co-Registration Skull Stripping Resampling N4 Bias Correction	Python	8327	First order, GLCM, GLSZM, GLRLM, NGTDM, GLDM, Deep features.	13 F selection (RelieF, GINI, CHSQ…) and 12 ML methods (CNN, SVM, RF, DT…)	VD = 76	2-Classes: AUC = 0.66 Accuracy = 64% 3-Classes: AUC = 0.58 Accuracy = 38%
37	Bakas [50] 2020	101	No	T1 T1C FLAIR T2 DTI DS-C	Automatic Enhancing Tumor Non/Enhancing Tumor Edema	Co-Registration Resampling Noise Filtering Histogram Matching	CaPTk	1612	First order, Volumetric, Wavelet, GLCM, GLRLM, GLSZM, NGTDM, Spatial information, diffusion properties	Forward Selection, SVM	5-FoldCV	Accuracy Advanced MRI = 73% Basic MRI = 74.3% KM
38	Park [51] 2020	216	No	T1C FLAIR DWI DS-C	Semiautomatic Enhancing tumor	Co-Registration Intensity Normalization Resampling	MATLAB	1618	First order, GLCM, GLRLM, Wavelet	LASSO	VD = 58	C-index = 0.64 KM Nomogram
39	Lu [52] 2020	181	No	T1C	Semiautomatic Whole tumor Enhancing tumor Nonenhancing tumor Necrosis	Intensity Normalization N4 Bias Correction	Python	333	Shape, First order, GLCM, GLDM, GLRLM, GLSZM, NGTDM VASARI	VHA, RFS	VD = 78	AUC = 0. 96 C-index = 0.90
40	Baid [53] 2020	346	Yes BRATS	T1 T1C FLAIR T2	Automatic. Whole Tumor Enhancing tumor Tumor core	Co-Registration N4 Bias Correction Normalization	MATLAB	678	First order, Wavelet decomposition, GLCM	SpCoR, RF	VD = 53	Accuracy = 57.1%
41	Moradmand [11] 2021	260	Yes TCGA IVY	N/A	N/A	N/A	Python	N/A	Clinical, Tumor Data, PostSurgical Treatment, Molecular variables	CoxPH, RF, NN	TD = 78	C-index = 0.823 Bayesian Hyperparameter Optimization
42	Yan [8] 2021	688	Yes TCIA CGGA Local	DTI T2 Flair	Manual Whole Tumor	Coregistration Standardization	Python	N/A	Radiogenomics Clinical	CNN	VD = 77	C-index = 0.825 (VD-C-index = 0.79)
43	Priya [54] 2021	85	No	T1C	Manual Whole Tumor	N/A	TexRAD	36	Texture, Age	SVM NN RT	5-FoldCV	AUC = 0.811 Accuracy = 67% AUC CV = 0.71
44	Cepeda [55] 2022	203	Yes TCIA = 34 BraTS = 119	T1 T1C FLAIR T2	Hybrid (GLISTRboost) Enhancing tumor Nonenhancing tumor Edema	Re-Orientation Co-Registration Resampling Normalization	CaPTk	15,720	First Order, Histogram, Volumetric, Morphologic, GLCM, GLDM, GLRLM, GLSZM, NGTDM	Gini Index, FCBF, InfoGain / LR, NB, kNN, RF, SVM, NN	TD = 60	AUC = 0.98 Accuracy = 94% (TD-AUC = 0.77 TD-Accuracy = 80%) Naïve Bayes
45	Ben Ahmed [14] 2022	163	Yes BRATS	T1C	Automatic Enhancing tumor Tumor core Whole Tumor	Null-Voxel Reduction Data Augmentation 2D Transformation	Python	35,709	Snapshot	CNN	VD = 46	Accuracy = 74%
46	Ruan [56] 2022	200	Yes TCGA = 129	T1C T1 T2FLAIR	Manual Whole Tumor	Standardization	MATLAB 3D Slicer	665	First Order, VASARI, GLCM, GLDM, GLRLM, GLSZM, NGTDM	LASSO	VD	Radiomics C-Index = 0.935 RadiomicsVASARI C-Index = 0.622

AUC: Area Under the Curve from Receiver Operating Characteristics; BraTS: Brain Tumor Segmentation Challenge Dataset; C: Contrast Enhanced; CGGA: Chinese Glioma Genome Atlas; CNN: Convolutional Neural Networks; CoxPH: Cox proportional hazards; CV: Cross Validation; DA: Discriminated Analysis; DS: Dynamic Susceptibility; DT: Decision Trees; DTI: Diffusion Tensor; DWI: Diffusion Weighted Image; Imaging; F: Radiomic Features; GBDT: Gradient Boosting Decision Tree; GLCM: Gray Level Co-Occurrence Matrix; GLDM: Gray Level Difference Matrix; GLSZM: Gray Level Size Zone Matrix; HR: Hazard Ratio;IVY: Ivy Glioblastoma Atlas Project; JIM: Joint Intensity Matrix; KM: Kaplan-Meier Survival Curves; kNN: K Nearest Neighbor; LASSO: Least Absolute Shrinkage and Selection operator; LOOCV: Leave One Out Cross Validation; LoG: Laplacian of Gaussian; LR: Logistic Regression; MAD: Median Absolute Deviation; mRMR: minimum Redundancy Maximum Relevance; N: Number of patients; NB: Naïve Bayesian; NGTDM: Neighborhood gray tobe difference matrix; NN: Neural Networks; PearsonC: Pearson’s Co-Relation Coeficient; RS-F-MRI: Resting State functional MRI; RF: Random Forest; RFE: Recursive feature elimination; RFS: Recursive Feature Selection; RT: Regression Trees; SPCA: Sparse Principal Component Analysis; SpCoR: Spearman’s Co-Relation; SFTA: Segmentation Fractal Texture Analysis); SVM: Support Vector Machine; TCGA: The Cancer Genome Atlas; TCIA: The Cancer Imaging Archives; TD: Testing Dataset; VD: Validation Dataset; VHA: Variable Hunting Algorithm; WM: White Matter. * (If all cases were from a public dataset, the number is not disclosed again).

**Table 2 medicina-58-01746-t002:** Itemized score for the Radiomics Quality Score of the included radiomics studies assessing survival prognosis in high grade gliomas.

**Author and Year**	**Yang** **2015**	**Lee** **2016**	**Kickingereder** **2016**		**Macyszyn 2016**	**Chaddad** **2016**	**Lao** **2017**	**Liu** **2017**	**Li** **2017**	**Ingrisch** **2017**	**Prasanna** **2017**		**Bae** **2018**	**Sanghani** **2018**	**Liao** **2018**	**Chaddad** **2018**	**Liu** **2018**	**Choi** **2019**
**Image protocol quality**	1	0	1		2	1	1	1	1	1	1		1	0	0	0	1	1
**Multiple segmentations**	0	0	1		0	0	1	1	1	0	0		1	0	0	1	1	1
**Phantom study on all scanners**	0	0	0		0	0	1	0	1	0	0		0	0	0	0	0	0
**Imaging at multiple time points**	0	0	0		0	0	0	0	0	0	0		0	0	0	0	0	0
**Feature reduction or adjustment for multiple testing**	−3	3	3		3	−3	3	3	3	3	3		3	3	3	−3	3	−3
**Multivariate analysis with nonradiomic features**	0	1	1		1	0	1	0	1	1	1		1	0	1	0	0	1
**Detect and discuss biologic correlates**	1	1	1		1	1	1	1	1	1	1		1	1	1	1	1	1
**Cutoff analysis**	0	1	1		1	1	1	0	1	0	0		1	0	0	1	0	0
**Discrimination statistics**	1	1	1		1	1	1	1	1	0	0		2	0	1	1	1	0
**Calibration statistics**	1	0	0		0	0	1	0	0	0	0		0	0	0	0	0	0
**Prospective study registered in trial data base**	0	0	0		1	0	0	0	0	0	0		0	0	0	0	0	0
**Validation**	2	−5	2		2	2	3	2	2	2	2		2	−5	2	2	2	2
**Comparison with criterion standard**	0	2	2		0	0	2	0	2	2	2		2	0	0	0	0	2
**Potential clinical utility**	0	0	0		0	0	0	0	0	0	0		0	0	0	0	0	0
**Cost-effectiveness analysis**	0	0	0		0	0	0	0	0	0	0		0	0	0	0	0	0
**Open science and data**	1	1	0		0	1	1	1	1	0	1		0	2	0	1	1	1
**Total points**	4	5	13		12	4	19	10	15	10	11		14	1	8	4	10	6
**% RQS**	11%	14%	36%		33%	11%	53%	28%	42%	28%	31%		39%	3%	22%	11%	28%	17%
**Author and Year**	**Tixier** **2019**	**Shboul** **2019**	**Chaddad** **2019**	**Chen** **2019**	**Kim** **2019**	**Chang** **2019**	**Osman** **2019**	**Chaddad** **2019**		**Um** **2019**	**Zhang** **2019**	**Han** **2020**	**Zhang** **2020**	**Suter** **2020**	**Bakas** **2020**	**Park** **2020**	**Cepeda** **2021**	**Ruan** **2022**
**Image protocol quality**	1	0	0	1	1	0	1	0		1	1	2	1	0	1	1	1	1
**Multiple segmentations**	1	1	0	1	1	1	1	1		1	1	0	1	1	1	1	0	1
**Phantom study on all scanners**	0	0	0	0	0	0	0	0		1	0	0	0	1	0	0	0	0
**Imaging at multiple time points**	0	0	0	0	0	1	0	0		0	0	0	0	0	0	0	0	0
**Feature reduction or adjustment for multiple testing**	3	3	−3	3	3	3	3	3		3	3	3	3	3	3	3	3	3
**Multivariate analysis with nonradiomic features**	0	0	1	1	1	0	1	1		0	1	0	1	0	0	1	1	1
**Detect and discuss biologic correlates**	1	1	1	1	1	0	1	1		1	1	1	1	1	1	1	0	1
**Cutoff analysis**	1	0	0	1	1	1	1	1		1	1	1	0	1	0	1	1	1
**Discrimination statistics**	0	1	1	1	1	1	1	1		0	1	1	1	1	1	1	1	1
**Calibration statistics**	0	0	0	0	0	0	0	0		0	1	0	0	0	0	0	1	0
**Prospective study registered in trial data base**	0	0	0	0	0	0	0	0		0	0	0	0	0	0	0	0	0
**Validation**	2	2	2	2	2	2	2	2		2	2	0	2	2	2	3	3	2
**Comparison with criterion standard**	0	0	2	2	2	1	2	2		0	2	2	2	0	0	2	2	1
**Potential clinical utility**	0	0	0	0	0	0	0	0		0	0	0	0	0	0	0	0	0
**Cost-effectiveness analysis**	0	0	0	0	0	0	0	0		0	0	0	0	0	0	0	0	0
**Open science and data**	1	2	1	1	0	0	3	1		0	1	1	1	1	2	1	2	1
**Total points**	10	10	5	14	13	10	16	13		10	15	11	13	11	11	15	15	13
**% RQS**	28%	28%	14%	39%	36%	28%	44%	36%		28%	42%	31%	36%	31%	31%	42%	42%	36%

## Data Availability

Not applicable.

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
