# Peer review of "Current Evidence, Limitations and Future Challenges of Survival Prediction for Glioblastoma Based on Advanced Noninvasive Methods: A Narrative Review"

_medicina, 2022, doi:10.3390/medicina58121746_

Round 1
Reviewer 1 Report
Although the topic of data analysis and prediction with AI and Machine Learning is a hot topic in the field of medicine in general and is of particular interest in the field of oncology, this review presents a topic that has already been partially exposed in the recent article by Zhu and colleagues (https://doi.org/10.3389/fonc.2022.924245).
Nevertheless, this work has excellent potential and could be improved.
Also, some writing/formatting errors are present:
- line 16: "tittle";
- Page 9: a blank page is present;
- line 169: the acronym DTI should also be explained in the text;
- line 200: an unnecessary conjunction "and" is present.
It might also be helpful to add additional diagrams or pictures that can easily explain the topic to the reader.
Author Response
We acknowledge and appreciate the thorough report issued by the reviewer. We believe his/her suggestions will improve the quality of the manuscript.
The use of english has been reviewed and edited according to reviewer´s suggestions.
We have created two new figures to illustrate the conventional workflow employed in studies implementing Machine Learning and Neural Networks, in order to illustrate the key steps of these investigations.
A modified version of the manuscript has been uploaded.
Reviewer 2 Report
The aim of the paper was to investigate the usefulness of radiomics and artificial intelligence in estimating overall survival for patients with glioblastoma. Since glioblastoma is a rare disease and since it is not possible for one clinical center to create a large dataset, the authors decided to review PubMed database in order to describe implementation of novel non-invasive technologies to provide an accurate survival estimation.
The authors included 46 studies into their review. Table 1 clearly summarized the results of those studies. Table 2 presented the radiomics quality score for articles whose methodology was based on radiomics. The discussion of this review shows clearly many limitations, both for creating such a review and for implementing artificial intelligence algorithms to survival prediction. The major limitations of previous publications include for example differences in patient selection and degree of tumour resectability, variability in preprocessing MRI scans or discrepancies in predicting survival in glioblastoma patients. The main challenge to use the described methods in a clinical context is that at present they are costly, time-consuming and require leaving care provision from clinical professionals. The authors of the review state it clearly that a culture of data sharing is needed.
The authors chose to review advanced non-invasive methods as predictors of overall survival in glioblastoma patients and completed the topic of this review.
The manuscript is clear, relevant for the filed and presented in a well-structured manner. The figure and tables properly show the data and are easy to interpret. The cited references are relevant. The conclusions are coherent and well supported by the listed citations.
My specific comments:
I would reconsider the title of the review so that it includes the word “limitation” since in my opinion this is the most significant message of the review.
Author Response
We appreciate the report submitted by the reviewer. The reviewer has conducted a thorough evaluation of our manuscript and highlighted the key messages we have found in our investigation.
We have reviewed the manuscript and amended it according to a proper use of English.
We have reconsidered the title of the review and changed it as follows: ¨Current evidence, limitations and future challenges of survival prediction for glioblastoma based on advanced noninvasive methods: A narrative review.¨
Round 2
Reviewer 1 Report
After revision this work has improved, I have only one last comment for the authors: you should improve the quality of Figure 2, as it has low quality and definition, and this does not allow a good reading of the explanations inside.
Author Response
Thank you for your review and suggestions. We have resized the images and embedded text.
We hope the new image will satisfy your demands.